# Effectiveness of a Sexuality Workshop for Nurse Aides in Long-Term Care Facilities

**DOI:** 10.3390/ijerph182312372

**Published:** 2021-11-24

**Authors:** Man-Hua Yang, Shu-Ting Yang, Tze-Fang Wang, Li-Chun Chang

**Affiliations:** 1College of Nursing, National Yang Ming Chiao Tung University, NO. 155, Sec. 2, Li-Nong Street, Beitou Dist., Taipei 112, Taiwan; mhyang@nycu.edu.tw; 2Taipei City Haoran Senior Citizens’ Home, Taipei 110204, Taiwan; ugjwu@hotmail.com; 3Department of Nursing, Chang Gung University of Science and Technology, Taoyuan 333, Taiwan; lichunc@mail.cgust.edu.tw

**Keywords:** long-term care facilities, sexuality workshop, knowledge of sexuality, attitude toward sexuality, quality of sexual life

## Abstract

Background: In long-term care facilities, there are frequent conflicts related to elderly residents’ sexual expression. Nurse aides usually handle such conflicts with negative or negligent attitudes; therefore, elderly sexuality is considered “problem behavior” and is stigmatized. Objectives: This study aimed to improve elderly residents’ quality of sexual life by enhancing nurse aides’ knowledge and attitudes toward elderly sexuality through sexuality workshops. Methods: A quasi-experimental study was conducted with 64 nurse aides and 58 residents, who were divided into two groups, i.e., an experimental group and a control group, according to the floor where the residents resided. The nurse aides in the experimental group participated in sexuality workshops and were compared with those in the control group with respect to their knowledge of and attitudes toward sexuality; the residents’ quality of sexual life was also compared between groups. Results: Compared with the control group, in the experimental group, the nurse aides’ knowledge of and attitudes toward elderly sexuality as well as the residents’ quality of sexual life significantly and continually improved after the sexuality workshops. Conclusion: The four-week sexuality workshop is effective and may be used as an example in developing occupational education programs regarding elderly sexuality in long-term care facilities.

## 1. Introduction

Taiwan is an aging society. Owing to its aging population, the demand for long-term care (LTC) facilities is increasing. In the past, the sexual expression of elderly residents in care facilities has usually been ignored or even stigmatized; however, many residents remain highly interested in sex but are deprived of the free expression of their sexuality [1,2,3]. In addition, there are many stereotypes and misunderstandings that are prevalent in society about the sex lives of seniors. Therefore, elderly sexuality is often neglected or negatively handled in LTC facilities, and no measures have been developed or undertaken to address this issue [4,5].

If nurse aides, as first-line caregivers in LTC facilities, have inadequate knowledge and negative attitudes toward elderly sexuality, elderly residents’ expression of their sexuality is discouraged, resulting in conflicts of different magnitudes between the two parties [5]. Therefore, this study aims to improve knowledge and attitudes toward elderly sexuality among nurse aides, increase their capability to handle issues related to elderly sexuality, and in turn improve the quality of sexual life of elderly residents and help them achieve sexual satisfaction.

### 1.1. Sexuality of Elderly Residents

Sex has physiological, psychological, and social dimensions that interact with each other. An obvious decline in sexual function occurs in elderly individuals not only due to a decrease in physiological functions but also due to diseases that impair their sensory functions and obstruct sexual signal conduction, thereby leading to sexual dysfunction [6,7]. Physiologically, sexual dysfunction is associated with multiple chronic diseases and the side effects of the administration of multiple drugs, such as antihypertensive agents, cardiovascular drugs, antidepressants, tranquilizers, antiulcer drugs, and diuretics [1,8]. Psychological and social factors preventing elderly individuals from engaging in sexual intimacy include the absence of positive social policies, lack of sexuality studies, absence of partners, negative portrayal in the media, negative psychological factors, and difficulty in discussing sexuality with medical care personnel; these factors all contribute to the negligence of elderly sexuality [9].

Residents’ sexual expression is often the subject of disdain and derision by LTC facility staff because of their insufficient sexual knowledge, resulting in elderly residents’ embarrassment and fear of expressing their sexual needs [10]. Satisfaction of the needs for intimate relationships favors a physiological–psychological balance and helps avoid feelings of early debility and unnecessary sexual fantasy; thus, such satisfaction is an important contributor to the quality of life of elderly individuals.

### 1.2. Nurse Aides’ Attitudes toward Elderly Sexual Expression

Nurse aides are the main care providers in LTC facilities; therefore, their gerontological knowledge and attitudes affect the quality of life of elderly residents [11]. However, nurse aides encounter considerable challenges in dealing with “sexual issues” because they are not trained on this subject.

In a survey on care for elderly sexuality in LTC facilities by the American Medical Directors Association in September 2013, it was found that only 23.3% of facilities had established policies about the sexual behavior of elderly residents, and only 13.4% of caring staff received education on sexuality. This survey indicated that many facilities had no clear policies to deal with these issues and lacked sufficient knowledge, which resulted in misunderstandings and prejudice among nurse aides regarding elderly sexuality; therefore, elderly residents’ sexual expression had been severely limited [12,13]. Sexual behaviors are not included in the routine assessment of LTC facilities [5,9], and staff members often conceal residents’ expression of sexuality and intimate relationships because they consider sexuality to not be one of the necessary physical functions to maintain the body and to be beyond their care responsibility [1,12,14,15].

Furthermore, society holds a negative attitude toward elderly sexuality. Although facility staff are aware of the sexual desires of elderly residents, they face tremendous pressure imposed by cultural values, ethics, individual religious beliefs, and inadequate training and education on sexual health [16]. Facility staff experience discomfort, panic, or denial when encountered with residents’ sexual expression; they feel unpleasant and embarrassed when discussing such behaviors with coworkers and even consider or label sexual expression as a “problem behavior” [17]. Therefore, sexual behaviors are discouraged in facilities because of concerns that it may harass or harm other residents or staff [5,18].

### 1.3. Sexuality Education for Nurse Aides

Only with accurate knowledge and positive attitudes toward sexuality can nurse aides properly handle issues with sexual expression among elderly residents, which in turn can improve the quality of sexual life of elderly residents [8,19] and reduce conflicts between nurse aides and residents with sexual needs when handling these issues.

In Taiwan, sexual education is mainly provided for preschool, school-age, and adolescent students [20], but none is provided for adults and elderly individuals, resulting in little public awareness of elderly sexuality. In addition, training in LTC facilities focuses only on the prevention of sexual harassment and rarely on elderly sexuality. Moreover, society generally has a conservative attitude toward sex; therefore, study subjects usually answer questions considering public perception rather than their own perceptions, thereby increasing study difficulty and decreasing study reliability owing to the lack of accurate and available data [21]. Medical providers have concerns about their professional role being influenced or their values being imposed on individuals inadvertently, causing residents to feel frustrated or guilty regarding their sexual needs. Previous studies have found that sexuality education programs can promote a change in attitude. The more sexual knowledge one receives, the more he or she feels comfortable and accepting when providing individual sexuality consultations and measures [8]. In a study evaluating the effects of an educational intervention on attitudes and beliefs toward sexuality in 112 nurse aides from LTC facilities [22], a three-hour educational program covering sexual behaviors, dementia, and legal issues was conducted. The results showed that the educational intervention improved knowledge on elderly sexuality and changed the participants’ previous negative attitudes toward residents’ sexual expression.

## 2. Material and Methods

### 2.1. Study Site and Recruitment

We selected a large LTC facility in Taipei as the study site. After approval from this facility was obtained, participant recruitment was initiated by posting an advertisement on bulletin boards at the facility from 27 April 2020 to 20 May 2020. To encourage nurse aides to participate in the study, they were provided with in-service education certification for the hours of the education workshop. The workshop was conducted in the form of a group discussion for two hours per week for a total of four weeks.

### 2.2. Inclusion and Exclusion Criteria

Nurse aides and residents with the intention to participate in the study were selected by purposive sampling. For nurse aides, the inclusion criteria were as follows: having >3 months of working experience as a nurse aide, being willing to participate in the study, and being capable of completing questionnaires by themselves or after explanations in Mandarin or Taiwanese. The exclusion criteria were as follows: individuals who planned to resign or could not complete all four sessions of the workshop during the duration of the study.

For residents, the inclusion criteria were as follows: having normal cognitive function, with the Short Portable Mental Status Questionnaire score of >8; being capable of communicating in Mandarin or Taiwanese; being willing to participate in the study; being capable of completing questionnaires by themselves or making selections on the questionnaires after explanation in Mandarin or Taiwanese; and having lived in the facility for >3 consecutive months. The exclusion criteria were as follows: individuals who were hospitalized during the study, planned to be discharged during the study, or had dementia.

### 2.3. Sample Size

In this study, the sample size was estimated to be 52 subjects using the software G-Power 3.1.9.2, with F tests—ANOVA: repeat measures, a mean of 341.2 and an effect size of 0.285; both values were based on the study by Lin and Lin [23]. A power of 0.8 (α = 0.05) was used. Considering an attrition rate of 20%, it was estimated that 65 subjects should be enrolled.

### 2.4. Study Design

A quasi-experimental design was adopted in this study with repeated measurements pre- and postintervention and four weeks after the intervention. The subjects were grouped by drawing lots according to the floor where the residents resided (e.g., control group was on first and second floor, experimental group was on third and fourth floor). The nurse aides participating in the workshop and the residents cared for by them were included in the experimental group, and the remaining residents were in the control group; both groups of residents lived in the facility as usual without other interventions. The questionnaires (anonymous) were distributed and collected from the nurse aides by their leaders and from the residents by the investigator. Each questionnaire was placed into an envelope to prevent the disclosure of information and to protect the privacy of the subjects.

### 2.5. Instruments

#### 2.5.1. Sexuality Workshop

According to Bandura’s theory of learning in the social environment [24], the learning strategy of the sexuality workshop was to increase knowledge, clarify mistakes, and enhance positive attitudes toward sex through observation and discussion. The program was designed by the research team by referring to the literature on long-term care and interviewing three veteran nurse aides regarding their opinions on elderly sexuality; three sexual education experts reviewed the process for the appropriateness of each topic and objective using a Likert scale. The scores given by the experts were between 4 (agree) and 5 (strongly agree).

The program covered four topics, i.e., “sexual physiology of elderly residents,” “sexual psychology of elderly residents,” “social aspects of the sexuality of elderly residents,” and “sexual ethics in facilities,” with one two-hour session per week for four weeks.

#### 2.5.2. Sexual Knowledge and Attitude Scale for Nurse Aides (SKAS-NA) in Long-Term Care Facilities

With reference to the Chinese version of the Aging and Sexuality Knowledge and Attitudes Scale (ASKAS) [25], the Sexual Knowledge and Attitudes Scale [26], and the Questionnaire on Sexual Attitudes and Job Satisfaction of Day Care Nurse Aides [27], the SKAS-NA was reconstructed for this study. The scale comprised questions on sexual knowledge and attitudes. Sexual knowledge contained three dimensions, i.e., “sexual physiology,” “sexual psychology,” and “sexual society.” It was a true–false item scored from 0 to 30; the higher the score, the greater the awareness about sexual knowledge. Sexual attitudes contained four dimensions, i.e., “sexual physiology,” “sexual psychology,” “sexual society,” and “sexual education”. Eighteen questions on sexual attitudes were scored from 18 to 90 on a Likert scale; the higher the score, the more positive the sexual attitude.

Five domestic experts were invited to analyze the validity of the SKAS-NA: a psychiatrist, the professor of health education, the professors of gerontological nursing, and the specialist of sexual education. By evaluating the appropriateness of each question using a Likert scale, the appropriateness was evaluated for each question with a score of 1–5. The higher the score was, the more appropriate the question. The content validity index (CVI) of the SKAS-NA was calculated to be 0.95. The calculated Cronbach’s α was 0.86 for sexual knowledge and 0.85 for sexual attitudes, with high consistency [28], thus indicating the favorable reliability of the tool.

#### 2.5.3. Quality of Sexual Life Questionnaire for Residents (QSL-R) in Long-Term Care Facilities

With reference to the brief Taiwan version of the WHO Quality of Life Instrument (WHOQOL-BREF) [29] and the Quality of Sexual Life Scale [30], the QSL-R was developed for this study. The scale comprised questions on personal information as the first part and question on quality of sexual life as the second part. The second part contained four dimensions, i.e., “sexual physiology,” “sexual psychology,” “sexual society,” and “integrity,” which included 12 questions scored on a Likert scale, with a total score ranging from 12 to 60; the higher the score, the better the quality of sexual life.

Five domestic experts were invited to analyze the validity of the QSL-R by evaluating the appropriateness of each question using a Likert scale from 1–5. The higher the score was, the more appropriate the question. The calculated CVI of the QSL-R was 0.96. The calculated Cronbach’s α of the scale was 0.78, with high consistency [28], thereby indicating the favorable reliability of the tool.

### 2.6. Analysis of Data

The data were analyzed using SPSS version 22.0 (IBM Corporation, Armonk, NY, USA). Descriptive statistics (including the percentage, mean, and standard deviation) were adopted for analysis of the general data of the subjects. Independent samples *t*-tests and chi-square tests were used to compare the differences in the general data between the experimental and control groups. A generalized estimating equation was adopted to test for any significant differences in sexual knowledge, sexual attitudes, and quality of sexual life at pre- and postintervention and four weeks after the intervention between the experimental and control groups.

### 2.7. Ethical Considerations

The subjects who provided an informed consent form could withdraw from the study at any time with no violation of any of their rights or interests.

## 3. Results

### 3.1. Analysis of Demographic Data of the Subjects

In this study, nurse aides and residents from a large long-term care facility in Taipei were enrolled from 24 May 2020 to 30 July 2020, by purposive sampling (subjects residing on the first and second floors were included in the control group and those on the third and fifth floors were included in the experimental group). A total of 73 nurse aides met the inclusion criteria, and 64 nurse aides completed the study, with 31 in the experimental group and 33 in the control group. A total of 149 residents met the inclusion criteria, and 100 of them agreed to participate in the study, with 68 subjects assigned to the experimental group and 32 subjects assigned to the control group. The attrition rates of nurse aides and residents were 5.9% and 12.1%, respectively. Because of the great difference in the number of residents enrolled between the two groups, samples were paired by the predictive factors that affect the quality of sexual life, including sex, age, education background, religious belief, and ethnicity, which were determined by literature review. Finally, 29 residents completed the study in each group. Refer to Figure 1 for details.

#### 3.1.1. Demographic Data of the Nurse Aides

Among the 62 subjects, 31 were in the experimental group and 33 were in the control group; the mean age was 54.9 years in the experimental group and 54.18 years in the control group. Both groups were predominantly female. Regarding educational background, the proportions of nurse aides in the experimental group who were middle school and high school graduates were both 35.5%, and high school graduates (30.0%) were dominant in the control group. Married participants were dominant in both groups. Overall, 75.8% (47 subjects) of the subjects had never received sexual education, and 41.9% (26 subjects) had experienced sexual harassment, mainly verbal harassment (Table 1). Independent samples *t*-tests and χ^2^-square tests were adopted to compare the difference in general information between the two groups. No statistically significant difference was observed in any item between the two groups (*p* > 0.05).

#### 3.1.2. Demographic Data of the Residents

The subjects were paired by factors, including sex, age, education background, religious beliefs, and ethnicity. In each group, there were 21 males and 8 females with primary school graduation, Buddhist beliefs, and Hokkien ethnicity. The mean ages were 77.45 and 78.86 years in the experimental and control groups, respectively (Table 2). Independent samples *t*-tests and χ^2^-square tests were adopted to compare the difference in general information between the two groups. No statistically significant difference was observed in any item between the two groups (*p* < 0.05).

### 3.2. Change in Sexual Knowledge Scores of the Nurse Aides

Sexual knowledge scores were compared between the groups pre- and postintervention and four weeks postintervention (Table 3). No significant difference was observed between the two groups (*β* = 3.05, *p* = 0.059) or between “posttest and four weeks postintervention” and preintervention (*p* > 0.05). In the analysis of the interaction between group and time point, the regression coefficients of “experimental group × posttest” (*β* = 6.17, *p* = 0.001) and “experimental group × four weeks posttest” (*β* = 5.89, *p* = 0.001) both reached significance, indicating that the sexual knowledge scores in the experimental group were 6.17 and 5.89 points higher than those in the control group at posttest and four weeks postintervention, respectively. Thus, sexual knowledge significantly improved in the experimental group after the sexuality workshop.

### 3.3. Change in Sexual Attitude Scores of the Nurse Aides

The sexual attitude scores of the nurse aides were compared between the two groups at pretest, posttest, and four weeks postintervention (Table 4). No significant difference was observed between the two groups (*β* = 3.38, *p* = 0.110) or between pretest and four weeks after the workshop (*β* = −3.03, *p* = 0.79). However, the posttest score was significantly lower (5.36 points) than the pretest score (*p* = 0.004). In the analysis of the interaction between groups and time points, the scores in the experimental group at posttest and four weeks postintervention were 13.49 (*p* = 0.000) and 13.64 (*p* = 0.000) points higher than those in the control group at pretest, respectively, with statistical significance, indicating that sexual attitudes in the experimental group were significantly improved.

### 3.4. Effect of the Sexuality Workshop on Residents’ Quality of Sexual Life

The scores of the residents’ quality of sexual life were compared between the two groups at pretest, posttest, and four weeks postintervention (Table 5). No significant difference was observed between the two groups (*β* = 83, *p* = 0.645) or between “posttest and four weeks postintervention” and “pretest” (*p* > 0.05). However, in the interaction of “group × time,” the scores of the residents’ quality of sexual life in the experimental group at posttest and four weeks postintervention were 11.48 and 10.00 points higher than those in the control group, respectively, with statistical significance (both *p* = 0.000), indicating that the residents’ quality of sexual life in the experimental group significantly improved after the sexuality workshop intervention.

Because of the large difference in the number of male (*n* = 21) and female (*n* = 8) subjects, the sex variable was introduced in the analysis (Table 6). The score of males was 9.30 points higher than that of females, with statistical significance (*p* = 0.000); the scores at posttest and four weeks postintervention were significantly higher than that at pretest, indicating improvement in the quality of sexual life over time; in the interaction of “group × time”, the result was consistent with that before the sex variable was introduced. All results showed that the residents’ quality of sexual life in the experimental group significantly improved after the sexuality workshop intervention. However, in the interaction of “sex × time”, compared with that of females at pretest, the quality of sexual life in males significantly decreased (*p* = 0.000) both at posttest and four weeks postintervention, indicating decreased quality of sexual life in males over time.

## 4. Discussion

In the investigation on knowledge and attitudes toward elderly sexuality in Pingtung by Huang et al. [31] and the study on elderly sexuality knowledge in nursing students by Zhu et al. [32], the average correctness rates on tests of elderly sexuality knowledge were 60% and 54.5%, respectively, showing a generally low correctness rate. In contrast, in this study, the average correctness rate on the test of elderly sexuality knowledge in the experimental group increased from 71% at pretest to 90.6% at posttest and to 93.6% at four weeks postintervention. In addition, the nurse aides’ misunderstanding and stereotypes of elderly sexuality were clarified during the sexuality workshop, and the participants expressed that they could better understand the sexual needs of elderly individuals.

In terms of sexual attitudes, the nurse aides’ average score in the experimental group increased from 66.29 at pretest to 74.42 at posttest and to 76.90 at four weeks postintervention, with an increase in the positive sexual attitude rate from 73.7% to 85.4%. In a study on the relationship between the sexual attitudes and job satisfaction of day care nurse aides by Hsieh [27], the nurse aides’ average sexual attitude score was 68.39 ± 9.13, with a positive sexual attitude rate of 75.9%. The sexual attitude score was similar in both studies during the pretest and became more positive after the intervention, thereby supporting the effectiveness of the sexuality workshop.

Compared with that at the pretest, the overall sexual attitude of all nurse aides significantly decreased posttest (*β* = −5.36, *p* = 0.004) and showed a nonsignificant decrease at four weeks postintervention (*β* = −3.03, *p* = 0.79), which may have been caused by the decrease in the sexual attitude scores in the control group from 62.91 at pretest to 57.55 at posttest and to 59.88 at four weeks postintervention, indicating negative sexual attitudes. It was deduced from the above results that owing to the lack of sexual knowledge, the nurse aides in the control group were incapable of properly dealing with the sexual expression and needs that always existed in the residents; thus, negative emotions, such as embarrassment, disgust, or discomfort, were elicited, which was reflected in the results of the questionnaire.

The results of this study indicate that attending sexuality workshops has a positive effect on the sexual attitudes of nurse aides, which is consistent with the conclusion derived from previous studies that sexual attitudes are improved with enhanced sexual knowledge [27,31,33,34]. Moreover, during the sharing and discussion section of the workshop, the nurse aides expressed that they could understand the existing sexual needs of the residents and have a positive attitude toward elderly sexuality after their sexual knowledge was improved. Therefore, it can be concluded that the sexuality workshop in the study had a positive effect on nurse aides’ sexual attitude and was recognized to be effective.

After the intervention, the residents’ quality of sexual life in the experimental group significantly improved at posttest and four weeks postintervention, indicating that correct sexual knowledge and positive attitudes toward elderly sexuality by nurse aides improved the residents’ quality of sexual life, which is similar to the results of the study by Bauer et al. [22]. Because of a large intragroup difference in the number of male and female residents, the sex variable was introduced in the analysis of the effect of sexuality workshops on the quality of sexual life. It was shown that the quality of sexual life in males was significantly higher than that in females; this could be because females are more likely to suppress their needs for sex under the influence of society and culture [31]. In the analysis of the interaction between sex and time point, it was shown that the quality of sexual life in males decreased over time, which may be due to the imbalanced male–female ratio or the lower quality of sexual life in females than in males. This further supports the results that the attitude toward sexuality in females is generally more conservative than that in males, and this attitude becomes even more conservative with age [31].

Last, the nurse aides’ sexual knowledge and attitude and the residents’ quality of sexual life in the experimental group were enhanced postintervention and remained so even four weeks after the intervention. This observation may be because the cases used in the program were real clinical cases that attracted the nurse aides’ interest to learn actively and attentively. The cases were observed so that the nurse aides could discuss and share opinions with each other in the classes proactively and enthusiastically, and in their spare time, the nurse aides took notes to share their viewpoints before and after the classes. During the study, when there were incidents related to elderly sexual needs, the nurse aides discussed positive and proper solutions, thereby decreasing their negative attitudes, such as negligence or chastising of residents. They gradually corrected their misunderstanding of elderly sexuality and became aware that elderly individuals remain interested in and have needs for sexuality, even though they live in care facilities and are aging. The nurse aids could have more positive attitudes toward residents’ sexual expression, thereby reducing conflicts between nurse aids and residents.

### Limitations

In the past, only a few studies have been conducted on sexual topics, which are highly private and easily influenced by society and culture. The authenticity of the questionnaires could not be ensured because subjects may have selected answers that were more accepted by the general public in their culture; actual thoughts can only be obtained if a familiar relationship is established with subjects or if the questionnaires are recommended by others. Therefore, it is suggested that the number of subjects who are interviewed and observed be increased to analyze the results and to improve the deficiency of quantitative studies through qualitative studies.

The test–retest reliability and construct validity were not built in SKAS-NA and QSL-R. It is a limitation in our study to validate the effect of the sexuality workshop.

## 5. Conclusions

It was shown that the nurse aides’ sexual knowledge and attitudes in the experimental group were superior to those in the control group after four sessions of a sexuality workshop intervention, and the effects persisted at four weeks postintervention. Similarly, the residents’ quality of sexual life significantly improved after the nurse aides participated in the sexuality workshop, and the effects remained even four weeks postintervention. It is recommended that topics on elderly sexuality be included during in-service training for long-term care staff so that elderly residents’ basic needs, which have long been neglected, are appropriately cared for and that the quality of sexual life in elderly residents can be improved.

## Figures and Tables

**Figure 1 ijerph-18-12372-f001:**
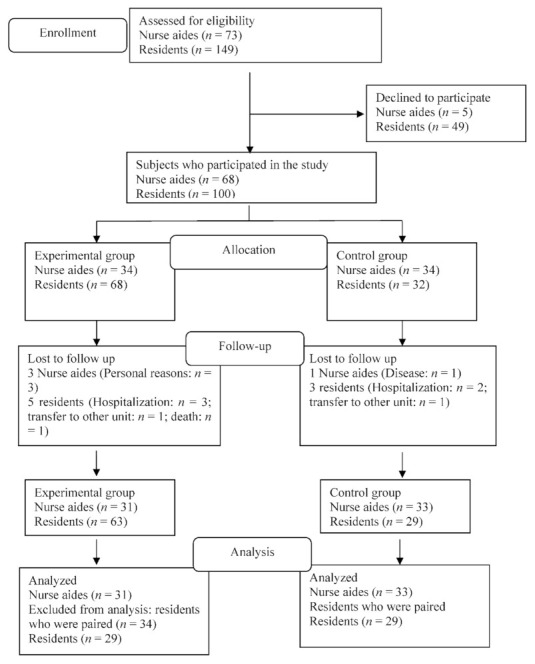
The flow chart of the subjects.

**Table 1 ijerph-18-12372-t001:** Demographic data of the nurse aides.

Item		Experimental Group (*n* = 31)	Control Group (*n* = 33)	*t*/χ^2^	*p*
		M ± SD, *n* (%)	M ± SD, *n* (%)		
Age		54.90 ± 7.42	54.18 ± 7.63	0.383	0.703
Working hours		264.38 ± 72.36	232.36 ± 59.45	1.94	0.57
Sex	Male	6 (19.4%)	7 (21.2%)	0.34	0.854
	Female	25 (80.6%)	26 (78.8%)		
Educational background	Primary school	5 (16.1%)	4 (12.1%)	3.17	0.544
	Middle school	11 (35.5%)	9 (27.3%)		
	High school	11 (35.5%)	10 (30.3%)		
	University	4 (12.9%)	9 (27.3%)		
	Master’s	0 (0%)	1 (3.0%)		
Marital status	Unmarried	2 (6.5%)	7 (21.2%)	2.94	0.41
	Married	14 (45.2%)	13 (39.4%)		
	Divorced	10 (32.3%)	8 (24.2%)		
	Widowed	5 (16.1%)	5 (15.2%)		
Fixed partner	None	16 (94.1%)	17 (85%)	2.77	0.13
	Yes	1 (5.9%%)	3 (15%)		
Religious beliefs	None	2 (6.5%)	9 (27.3%)	0.87	0.27
	Yes	29 (93.5%)	24 (72.7%)		
Ethnicity	Hokkien	19 (61.3%)	12 (36.4%)	6.54	0.72
	Hakka	0	3 (9.1%)		
	Aborigines	0	1 (3.0%)		
	New immigrants	12 (38.7%)	16 (48.5%)		
	Migrant workers	0	1 (3.0%)		
Nationality	Chinese	12 (100%)	15 (94.4%)	2.19	1
	Vietnamese	0	1 (5.6)		
Years of work experience	<1 year	4 (12.9%)	2 (6.1%)	3.11	0.72
	1–2 years	2 (6.5%)	3 (9.1%)		
	2–3 years	3 (9.7%)	3 (9.1%)		
	3–4 years	4 (12.9%)	5 (15.2%)		
	4–5 years	5 (16.1%)	2 (6.1%)		
	>5 years	13 (41.9%)	18 (54.5%)		
Sexual education	None	24 (77.4%)	23 (69.7%)	0.49	0.49
	Yes	7 (22.6%)	10 (30.3%)		
Heard of sexual harassment	None	9 (29%)	18 (54.5%)	4.27	0.39
	Yes	22 (71%)	10 (45.5%)		
Experienced sexual harassment	None	19 (61.3%)	19 (57.6%)	0.91	0.76
	Yes	12 (38.7%)	14 (42.4%)		
Type of sexual harassment	Verbal	10 (32.3%)	9 (27.3%)	0.19	0.66
	Physical	5 (16.1%)	9 (27.3%)	1.16	0.28
	Visual	2 (6.5%)	1 (3.0%)	4.19	0.61
Source of sexual knowledge	School teachers	13 (41.9%)	15 (45.5%)	0.8	0.78
	Friends/schoolmates	18 (58.1%)	8 (24.2%)	7.58	0.06
	TV	14 (45.2%)	9 (27.3%)	2.22	0.14
	Books and magazines	13 (41.9%)	10 (30.3%)	0.94	0.33

Note: *t*—*t*-test; χ^2^—chi-square test.

**Table 2 ijerph-18-12372-t002:** The demographic data of residents.

Item		Experimental Group (*n* = 29)	Control Group (*n* = 29)	*t*/χ^2^	*p*
		M ± SD	M ± SD		
Age		77. 45 ± 8.45	78.86 ± 7.86	−0.66	0.51
Sex	Male	21 (72.4%)	21 (72.4%)	0.00	1.00
	Female	8 (27.6%)	8 (27.6%)		
Educational background	Illiterate	2 (6.9%)	3 (10.3%)	1.65	0.98
	Primary school	15 (51.7%)	13 (44.8%)		
	Middle school	5 (17.2%)	5 (17.2%)		
	High school	4 (13.8%)	5 (17.2%)		
	University	2 (6.9%)	3 (10.3%)		
	Master’s	1 (3.4%)	0 (0%)		
Religious beliefs	None	6 (20.7%)	5 (17.2%)	6.35	0.25
	Buddhism	12 (41.4%)	10 (34.5%)		
	Taoism	3 (10.3%)	3 (10.3%)		
	General	7 (24.1%)	6 (20.7%)		
	Christianity	0 (0%)	5 (17.2%)		
	Others	1 (3.4%)	0 (0%)		
Ethnicity	Hokkien	18 (62.1%)	21 (72.4%)	0.96	0.87
	Hakka	3 (10.3%)	3 (10.3%)		
	Mainlanders	6 (20.7%)	4 (13.8%)		
	Others	2 (6.9%)	1 (3.4%)		
Chronic disease	Yes	29 (100%)	29 (100%)		
Taking medicines	Yes	29 (100%)	29 (100%)		
Smoking	No	16 (55.2%)	20 (69.0%)	1.89	0.64
	Yes	4 (13.8%)	3 (10.3%)		
	Given up	9 (31.0%)	6 (20.7%)		
Drinking	No	20 (69%)	25 (86.2%)	4.49	0.10
	Yes	1 (3.4%)	2 (6.9%)		
	Given up	8 (27.6%)	2 (6.9%)		
Sexual behaviors	Yes	13 (44.8%)	7 (24.1%)	2.75	0.10
	None	16 (55.2%)	22 (75.9%)		
Method of sexual behaviors	Fantasizing	13 (44.8%)	7 (24.1%)	2.75	0.10
	Touching	9 (31%)	4 (13.8%)	2.48	0.12
	Caressing	8 (27.6%)	4 (13.8%)	1.68	0.20
Reason for asexuality	Social perspective	9 (31%)	10 (34.5%)	0.08	0.78
	Disease	7 (24.1%)	9 (31%)	0.35	0.56
	Partner	6 (20.7%)	9 (31%)	0.81	0.37
Relationship status	None	63 (100%)	29 (100%)		
Wish to understand sexual education	No	19 (65.5%)	19 (65.5%)	0.00	1.00
	Yes	10 (34.5%)	10 (34.5%)		

Note: *t*—*t*-test; χ^2^—chi-square test.

**Table 3 ijerph-18-12372-t003:** Comparison of the nurse aides’ sexual knowledge between the two groups.

Variables	Pretest Mean ± SD	Posttest Mean ± SD	4 WKS Post Mean ± SD	*β* (95% CI)	*p*
Group					
Experimental group	21.35 ± 5.83	27.19 ± 3.06	28.10 ± 1.01	3.05 (−0.11–6.22)	0.059
Control group	18.30 ± 7.25	17.97 ± 2.80	19.15 ± 5.14	reference	
Time (Reference: Pretest)					
Posttest				−0.33 (−3.09–2.42)	0.813
4 WKS Post				0.85 (−1.91–3.61)	0.547
Group × Time (Reference: Control group × Pretest)					
Experimental group × Posttest				6.17 (2.55–9.79)	0.001 ***
Experimental group × 4 WKS Post				5.89 (2.50–9.28)	0.001 ***

Note: 4 WKS post—four weeks postintervention; *** *p* < 0.001.

**Table 4 ijerph-18-12372-t004:** Comparison of the nurse aides’ sexual attitudes between the two groups.

Variables	Pretest Mean ± SD	Posttest Mean ± SD	4 WKS Post Mean ± SD	*β* (95% CI)	*p*
Group					
Experimental group	66.29 ± 7.68	74.42 ± 8.54	76.90 ± 7.62	3.38 (−0.76–7.53)	0.110
Control group	62.91 ± 9.47	57.55 ± 3.89	59.88 ± 3.76	reference	
Time (Reference: Pretest)					
Posttest				−5.36 (−8.99 to −1.73)	0.004 *
4 WKS post				−3.03 (−6.41–0.35)	0.079
Group × Time (Reference: Control group × Pretest)					
Experimental group × Posttest				13.49 (7.92–19.06)	0.000 *
Experimental group × 4 WKS Post				13.64 (8.49–18.79)	0.000 *

Note: 4 WKS post—four weeks postintervention; * *p* < 0.05.

**Table 5 ijerph-18-12372-t005:** Comparison of the residents’ quality of sexual life between the two groups.

Variables	Pretest Mean ± SD	Posttest Mean ± SD	4 WKS Post Mean ± SD	*β* (95% CI)	*p*
Group					
Experimental group	34.97 ± 6.91	45.95 ± 3.09	45.71 ± 3.67	−0.83 (−4.34–2.69)	0.645
Control group	34.89 ± 6.32	35.52 ± 3.31	36.48 ± 2.94	reference	
Time (Reference: Pretest)					
Posttest				0.55 (−1.83–2.94)	0.651
4 WKS Post				1.51 (−1.17–4.21)	0.270
Group × Time (Reference: Control group × Pretest)					
Experimental group × Posttest				11.48 (7.57–15.39)	0.000 *
Experimental group × 4 WKS Post				10.00 (5.942–14.06)	0.000 *

Note: 4 WKS post—four weeks postintervention; * *p* < 0.05

**Table 6 ijerph-18-12372-t006:** Comparison of the residents’ quality of sexual life between the two groups with the sex variable introduced.

Variables	*β* (95% CI)	*p*
Sex (Reference: Female)	9.30 (6.49–12.13)	0.000 ***
Group (Reference: Control group)		
Experimental group	−0.83 (−3.33 to −1.67)	0.52
Time (Reference: Pretest)		
Posttest	6.39 (2.52–10.25)	0.001 **
4 WKS Post	8.06 (5.33–10.79)	0.000 ***
Group × Time (Reference: Control group × Pretest)		
Experimental group × Posttest	11.48 (8.03–14.93)	0.000 ***
Experimental group × 4 WKS Post	10.00 (6.52–13.48)	0.000 ***
Sex × group (Male × Experimental group)	−1.00 (−1.86–1.86)	1.00
Sex × Time (Reference: Female × Pretest)		
Male × Posttest	−8.05 (−11.81 to −4.30)	0.000 ***
Male × 4 WKS Post	−9.03 (−12.46 to −5.62)	0.000 ***

Note: 4 WKS postintervention—four weeks postintervention; ** *p* < 0.01, *** *p* < 0.001.

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
