# Peer review of "Effectiveness of a Sexuality Workshop for Nurse Aides in Long-Term Care Facilities"

_ijerph, 2021, doi:10.3390/ijerph182312372_

Round 1

Reviewer 1 Report

Dear authors,

Below there some comments, doubts, and minor and major concerns regarding your work.

----------------

-Were any difference between those who accepted to participate and those who declined.

-All the nurses caring for the residents of the floor which received the treatment, received the training? All these nurses were willing to receive this training?

* Sample size

-Can you please show me in detail how did sample size was estimated using G-Power, so that I can replicate it?

- Lin & Lin study used a graduate students sample. Please justify why did you base you sample size calculation in this study.

* Instruments

  • It should be explained why SKAS-NA was “reconstructed”. It should be also justified whether CVI and Alpha are enough evidence of the validation of the instrument, i.e., why no testing for test-retest reliability, cross-cultural validity, convergent, discriminant validity etc. If this cannot be considered a validated scale, it should also be a limitation.
  • The same applies to QSL-R

* Analyses and Results

-Why did you apply GEE instead of mixed-effect models?

-Table 1.  Please indicate what a fixed partner is. Should this apply only to those not married? If so, there should be 17 instead of 18 in the experimental group

-Table 3, 4, 5 and related results

Please format it properly to make it readable. Also, I find the labels confusing: What is the interaction between Experimental group × Post or  4 WKS? Should it be Group (Experimental vs control) instead of experimental group?

I do not understand how interaction was tested. There should one interaction per outcome (Group x time), so what are the other interactions? IF they are analyses dissected by time, I am afraid it is not the best way to test it.

- Table 6 and related results.

As you mention no previous aim regarding sex, this variable should only be taken into account as a control variable. Otherwise, this probably should be corrected for multiple comparison. But, more important, it seems to me incoherent to test sex “Because of the large difference in the number of male (n = 21) and female (n = 8)” as you State. On table 2 you tested whether both groups were balanced regarding sex. If you do not take these results into account on posterior analyses, then you control for every other demographic variable which is a potential confounding variable. But then, what would be the point of table 2.

Conclusion

Line -371: “It was demonstrated that the nurse aides’ sexual knowledge and attitudes in the experimental group”. I strongly disagree that your research design allows you to demonstrate this effect. Please relax language. Suggestion: showed instead of demonstrated.

Author Response

1.Were any difference between those who accepted to participate and those who declined.

A total of 73 nurse aides met the inclusion criteria, 5 people declined to join. Because they had private matters during the sexuality workshop.

2.All the nurses caring for the residents of the floor which received the treatment, received the training? All these nurses were willing to receive this training?

The subjects were grouped by drawing lots according to the floor where the residents resided. The nurse aides joined the workshop in the experimental group, and others didn’t reveive any treatment in the control group.

These nurse aides were willing to join the workshop, because they can’t deal with the difficult situation in elderly.

*3.Sample size

-Can you please show me in detail how did sample size was estimated using G-Power, so that I can replicate it?

In this study, the sample size was estimated to be 52 subjects using the software G-Power 3.1.9.2, with F tests- ANOVA : repeat measures, a mean of 341.2 and an effect size of 0.285; both values were based on the study by Lin and Lin [23]. A power of 0.8 (α = 0.05) was used. Considering an attrition rate of 20%, it was estimated that 65 subjects should be enrolled.

- Lin & Lin study used a graduate students sample. Please justify why did you base you sample size calculation in this study.

Our participants were nurse aides that rarely received sexual education in the previous stuies. Our study design was similar to Lin & Lin study. The intervention also was sexual education in Lin & Lin study. We took into account the paticipants were different, and decided the attrition rate was 20%.

*4. Instruments

  • It should be explained why SKAS-NA was “reconstructed”. It should be also justified whether CVI and Alpha are enough evidence of the validation of the instrument, i.e., why no testing for test-retest reliability, cross-cultural validity, convergent, discriminant validity etc. If this cannot be considered a validated scale, it should also be a limitation.
  • The same applies to QSL-R

We didn’t build up the test-retest reliability and construct validity in SKAS-NA and QSL-R. We add the limitation of scales in the manuscript.

* 5.Analyses and Results

-Why did you apply GEE instead of mixed-effect models?

GEE can be used to measure the continuous variable and categorical variable, and mixed-effect models can be used to measure the continuous variable. We added sex variable to measure the quality of sexual life, so GEE was suitable for our study.

6.Table 1.  Please indicate what a fixed partner is. Should this apply only to those not married? If so, there should be 17 instead of 18 in the experimental group

Fixed partner is a significant other in an intimate relationship. Fixed partner is applied to those not married, the number and percentage of fixed partner have been revised in Table 1.

  1. Table 3, 4, 5 and related results

Please format it properly to make it readable. Also, I find the labels confusing: What is the interaction between Experimental group × Post or  4 WKS? Should it be Group (Experimental vs control) instead of experimental group?

We have formatted the tables, and added the annotation of “ Group × Time” in Table 3,4,5.

I do not understand how interaction was tested. There should one interaction per outcome (Group x time), so what are the other interactions? IF they are analyses dissected by time, I am afraid it is not the best way to test it.

In “Group × Time”, “control group × pretest” was the reference variable, we could compare two times posttest of experimental group to the pretest of control group. The effect of experimental group over time has been seen.

8.Table 6 and related results.

As you mention no previous aim regarding sex, this variable should only be taken into account as a control variable. Otherwise, this probably should be corrected for multiple comparison. But, more important, it seems to me incoherent to test sex “Because of the large difference in the number of male (n = 21) and female (n = 8)” as you State. On table 2 you tested whether both groups were balanced regarding sex. If you do not take these results into account on posterior analyses, then you control for every other demographic variable which is a potential confounding variable. But then, what would be the point of table 2.

Sex variable was non-significant between two groups. The overall sex distribution of samples was male more than female. Gerder is an important factor of sexuality in literature review. We put the sex variable into the quality of sexual life. The result showed male’s quality of sexual life was different over time.

Conclusion

Line -371: “It was demonstrated that the nurse aides’ sexual knowledge and attitudes in the experimental group”. I strongly disagree that your research design allows you to demonstrate this effect. Please relax language. Suggestion: showed instead of demonstrated.

Thank you for your suggestion. We have used the “showed” instead of “demonstrated”.

Reviewer 2 Report

This is a very timely and interesting study assessing the effects of a short term sexuality workshop for nurse aides on the perceived quality of sexual life in elderly living in a care facility. The manuscript is clearly written and the study design is appropriate. In order for this manuscript to have a bigger impact, I would suggest the following:

  1. Provide access to the actual workshop or at least provide a way more detailed description of its design
  2. Provide access to both, the questionnaire aimed to assess the nurse's sexual knowledge and attitude and the quality of sexual Life questionnaire for Residents as well as a more thorough explanation of what each score means; it's hard to really understand how the conclusions are being reached
  3. Provide the names and/or backgrounds and/or credentials of the "experts" assessing the workshop and the questionnaires. Just mentioning that they are "experts" is not enough information to be able to assess their capabilities in the subject
  4. Present the data in figures. Tables are great for specifics of the data but are harder to interpret.

Author Response

  1. Provide access to the actual workshop or at least provide a way more detailed description of its design
  2. Provide access to both, the questionnaire aimed to assess the nurse's sexual knowledge and attitude and the quality of sexual Life questionnaire for Residents as well as a more thorough explanation of what each score means; it's hard to really understand how the conclusions are being reached
  3. Provide the names and/or backgrounds and/or credentials of the "experts" assessing the workshop and the questionnaires. Just mentioning that they are "experts" is not enough information to be able to assess their capabilities in the subject
  4. Present the data in figures. Tables are great for specifics of the data but are harder to interpret.
  1. The workshop covered four topics, i.e., “sexual physiology of elderly residents,” “sexual psychology of elderly residents,” “social aspects of the sexuality of elderly residents,” and “sexual ethics in facilities,” with 2 hours at a time, once a week for 4weeks. One topic was including the lecture with the researcher using power-point, video, pictures for 90 minutes, discussion for 20 minutes, and total summary for 10 minutes.
  2. (1)Sexual Knowledge and Attitude Scale for Nurse Aides (SKAS-NA)

The scale comprised questions on sexual knowledge and attitudes. Sexual knowledge contained three dimensions, ie., “sexual physiology,” “sexual psychology,” “sexual society,”. It was a true-false item scored from 0 to 30. Sexual attitudes contained four dimensions, ie., “sexual physiology,” “sexual psychology,” “sexual society,” “sexual education.” Eighteen questions on sexual attitudes were scored from 18 to 90 on a Likert scale.

(2) Quality of Sexual Life Questionnaire for Residents (QSL-R)

The scale comprised questions on personal information as the first part and question on quality of sexual life as the second part. The second part contained four dimensions, ie., “sexual physiology,” “sexual psychology,” “sexual society,” “integrity,”, included 12 questions scored on a Likert scale, with a total score ranging from 12–60.

  1. We invited five domestic experts to evaluate the workshop and the scales. They are a psychiatrist, the professor of health education, the professors of gerontological nursing, and the specialist of sexual education.
  2. Figure 1 is the flow-chart. The number of cases have been showed in figure 1. In order to enhance the table’s readablility, we have formatted the tables, and added the annotation of “ Group × Time” in Table 3,4,5.
